# Mimetic Initialization Helps State Space Models Learn to Recall

## Abstract

Recent work has shown that state space models such as Mamba are significantly worse than Transformers on recall-based tasks due to the fact that their state size is constant with respect to their input sequence length. But in practice, state space models have fairly large state sizes, and we conjecture that they should be able to perform much better at these tasks than previously reported. We investigate whether their poor copying and recall performance could be due in part to training difficulties rather than fundamental capacity constraints. Based on observations of their "attention" maps, we propose a structured initialization technique that allows state space layers to more readily mimic self-attention. Across a variety of architecture settings, our initialization makes it substantially easier for Mamba to learn to copy and do associative recall from scratch.

## 1 Introduction

State Space Models (SSMs) show promise as a potential replacement for Transformers (Vaswani, 2017) with substantially lower inference costs (Gu & Dao, 2023; Dao & Gu, 2024). While Transformer memory grows linearly with the input sequence length, SSMs use only a constant amount, compressing all the context into a fixed-size state. SSMs perform comparably to Transformers on a variety of common benchmarks. However, recent research has highlighted a set of tasks on which SSMs perform substantially worse than Transformers (Waleffe et al., 2024), particularly those involving copying or recall (Jelassi et al., 2024; Arora et al., 2024). This is perhaps unsurprising, as it is harder to recall from a compressed, fixed-size representation, particularly as its length grows.

Nevertheless, SSMs use relatively large state sizes in practice, and we wonder if their poor performance on tasks such as copying could be due to training difficulties rather than fundamental capacity constraints. We present a qualitative study of the failure modes of SSMs on the copying task. In particular, we inspect the time-dependent linear transformation matrix of Mamba layers, which is analogous to the attention map of self-attention layers. We compare these layers to their counterparts in self-attention/Mamba hybrid architectures that successfully learn to copy, and based on these comparisons, we propose a structured initialization technique that allows Mamba layers to more readily mimic self-attention. Our technique makes use of the fact that state space layers can be seen as a form of linear attention with a learnable, structured causal mask. We find evidence that such linear-attention-like Mamba layers arise naturally after large-scale pretraining, suggesting that this pattern may be fundamental to the recall abilities of SSMs.

**The proposed mimetic initialization allows Mamba to quickly learn to copy and do associative recall on up to $4\times$ longer strings**, and we show for the first time that **SSMs can achieve $2\times$ length generalization** or more. Mimetic initialization is **essentially compute-free**, but we show **it is comparable to pretraining** in allowing Mamba to learn to copy and recall. Our work helps to better understand the capacity of SSMs relative to Transformers in practice and can assist in further studies of their capabilities, which may have been underestimated by previous research.

**Related work** Recently, Jelassi et al. (2024) did a thorough investigation of the ability of state space models (in particular Mamba 1) to copy in comparison to Transformers. Their theoretical results demonstrate that SSMs with a fixed state size have fundamentally limited copying capacity, unlike Transformers which can strongly generalize. Empirically, they find that Transformers (especially with their proposed custom position embeddings) vastly outperform SSMs on copying, both

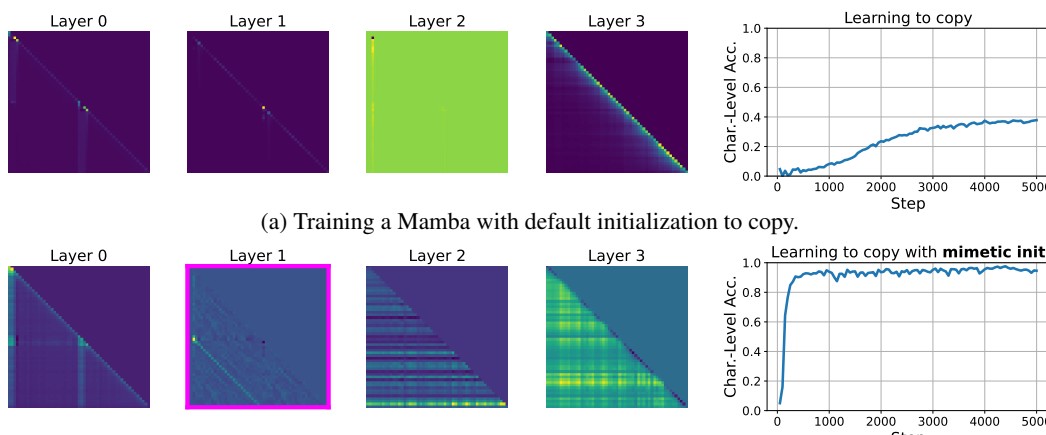

(a) Training a Mamba with default initialization to copy.

(b) Mamba with mimetic initialization learns to use its attention-like abilities.

Figure 1: Mambas initialized with our technique learn to copy more effectively than those with default initialization. We see evidence of copying ability in the Mamba attention maps; see Layer 1.

in terms of learning and length generalization. They note that in practice, SSMs may be better at copying than expected due to their relatively large state sizes, but do not observe very good copying performance in their experiments. Similarly, Arora et al. (2024) note that SSMs struggle on recall tasks due to their limited state size. They propose an effective intervention in the form of interleaved kernelized linear attention layers that boost recall performance. The second, improved version of the Mamba architecture improves upon associative recall ability, although the authors note that this task remains difficult for SSMs (Dao & Gu, 2024).

Initialization has been important for SSMs since their introduction to deep sequence modeling by Gu et al. (2021); a structured initialization of the state matrix was crucial to the performance of these earlier time-invariant SSMs (Gu et al., 2020; Gupta et al., 2022; Gu et al., 2022; Smith et al., 2023). Our work further demonstrates the importance of initialization for SSMs, taking inspiration from *mimetic initialization* (Trockman & Kolter, 2023; Trockman et al., 2022), which uses pretrained models as case studies of good initialization. For example, previous work noted that self-attention layers in pretrained Vision Transformers may try to imitate the local mixing ability of convolutions, which is reflected in the correlations between query/key and value/projection weights; initializing weights with statistical structure that mimics this pattern greatly improved trainability. We follow a similar methodology to propose a novel mimetic initialization technique for state space layers based on our observations that (1) these layers can represent linear attention, which can improve recall and (2) they sometimes approximate linear attention in pretrained models.

## 2 PRELIMINARIES

Recently, state space models have become popular as a choice of token mixing layer, i.e., as a replacement for self-attention. We refer to layers that use state space models for this purpose as *state space layers*. As it is common in the literature, with a slight abuse of definitions, we refer to architectures like Mamba 1 & 2 that use state space layers only for sequence mixing as *state space models*.

**State space models** For a scalar sequence $x \in \mathbb{R}^{\mathrm{T}}$, SSMs are linear recurrences of the form

$$h_{t+1} = \bar{A}h_t + \bar{B}x_t, \quad y_t = Ch_t, \tag{1}$$

where $h_t \in \mathbb{R}^{\mathrm{N}}$ is a hidden state, and $\bar{A} \in \mathbb{R}^{\mathrm{N} \times \mathrm{N}}$, $\bar{B} \in \mathbb{R}^{\mathrm{N} \times 1}$, $C \in \mathbb{R}^{1 \times \mathrm{N}}$ are the state space model parameters. The bar notation refers to the *discretized* form of the underlying parameters $A$ and $B$, as SSMs are traditionally continuous systems. Typically, some structure is imposed on $\bar{A} \in \mathbb{R}^{\mathrm{N} \times \mathrm{N}}$, such as diagonal-plus-low-rank (S4), diagonal (Mamba), or scalar-times-identity (Mamba 2).

In contrast, *selective* SSMs such as the S6 layer in Mamba allow the parameters $\bar{A}_t, \bar{B}_t, C_t$ to vary with time, i.e., depend on $x_t$. The particular state space layer in Mamba operates on sequences of

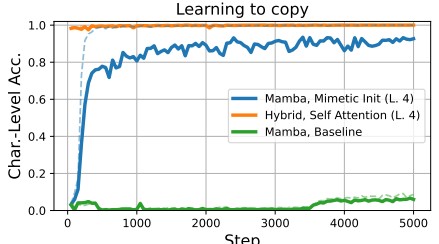 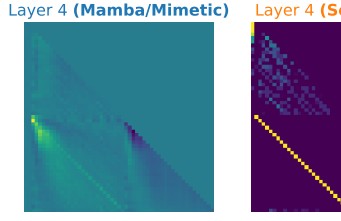

Figure 2: A hybrid Mamba architecture with one Self-Attention layer easily learns to copy. Dotted lines: performance on training length (50), solid: $2\times$ length generalization (100).

D-dimensional tokens $X \in \mathbb{R}^{\mathtt{D} \times \mathtt{T}}$. Indexing tokens with $t$ and *channels* with d, it computes

$$h_{(t+1),d} = \bar{A}_{td} h_{td} + \bar{B}_{td} X_{td}, \quad y_{td} = C_t h_{td}, \tag{2}$$

where $\bar{A}_{td}, \bar{B}_{td}, C_t$ depend on *all* channels of input $x_t$, but with different discretization parameters $\Delta_{td}$, hence the dependence of $\bar{A}_{td}$ and $\bar{B}_{td}$ on $d$. Define the underlying parameters $W_B, W_C \in \mathbb{R}^{\mathtt{N} \times \mathtt{D}}$, and $A \in \mathbb{R}^{\mathtt{D} \times \mathtt{N}}$. Let $W_\Delta \in \mathbb{R}^{\mathtt{D} \times \mathtt{D}}$ be a rank-$r$ matrix, and bias $b_\Delta \in \mathbb{R}^{\mathtt{D}}$. Then the continuous state space model parameters are computed as $B_t = W_B^T X_{:,t}$ and $C_t = W_C^T X_{:,t}$. The parameters of the discretized state space models are then computed as follows:

$$\Delta_{t,d} = \mathsf{softplus}(W_{\Delta d}^T X_{:,t} + b_{\Delta,d}), \quad \bar{A}_{td} = \exp(A_d \Delta_{t,d}), \quad \bar{B}_{td} = B_t \Delta_{t,d}. \tag{3}$$

Please refer to Dao & Gu (2024) for a more detailed discussion on selective SSMs.

**Matrix form of SSMs** The operations of Eq. 3 can be written concisely in matrix form:

$$\Delta \coloneqq \mathsf{softplus}\left(W_\Delta X + b_\Delta\right) \qquad \in \mathbb{R}^{\mathtt{D} \times \mathtt{T}} \tag{4}$$

$$\bar{B}_d \coloneqq W_B X \odot \mathbf{1}_n \Delta_d \qquad \in \mathbb{R}^{\mathtt{N} \times \mathtt{T}} \tag{5}$$

$$C \coloneqq W_C X \qquad \in \mathbb{R}^{\mathtt{N} \times \mathtt{T}} \tag{6}$$

$$\bar{A}_d \coloneqq \exp\left(A_d^T \Delta_d\right) \qquad \in \mathbb{R}^{\mathtt{N} \times \mathtt{T}} \tag{7}$$

As noted first by Ali et al. (2024), the time-varying discrete recurrence $h_{t+1} = \bar{A}_t h_t + \bar{B}_t x_t$, $y_t = C h_t$ can be unrolled and viewed as a matrix operation. Namely, channel $d$ of the output of an SSM layer, denoted with $Y_d \in \mathbb{R}^{\mathtt{T}}$, can be written as $Y_d \coloneqq M_d X$, where $M_d \in \mathbb{R}^{\mathtt{T} \times \mathtt{T}}$ is a matrix transformation dependent on $d$. Each matrix $M_d$ represents a time-dependent linear transformation, much like attention maps in self-attention. For $i, j \in [\mathtt{T}]$, the $M_d$ matrix of the Mamba state space layer for channel $d$ can be expressed as follows:

$$M_{d,i,j} = C_{:,i}^T \left(\Pi_{k=j+1}^i \mathsf{diag}(\bar{A}_{d,:,k})\right) \bar{B}_{d,:,j} \times \mathbf{1}\{i \le j\}. \tag{8}$$

As it will be useful later, we note that in practice, $A$ is parameterized as $A \coloneqq -\exp(A_{\mathsf{log}})$ with $A_{\mathsf{log}} \in \mathbb{R}^{\mathtt{D} \times \mathtt{N}}$. The selective state space layer of Mamba 2 is broadly similar to that of Mamba 1; it follows equations 4–7, but instead of having D different $A_d$ and $\Delta_d$, it has H independent $A$ and $\Delta$, each of which are repeated D/H times to construct $A_d$ and $\Delta_d$. Each of these H independent $A$ are parameterized as scalar-times-identity matrices, resulting in just $H$ parameters. These H components correspond to "heads", leading to only H unique $\bar{A}_d$ and $\bar{B}_d$ parameters, and only H "attention matrices" $M_d$ (*c.f.* Eq. 8), as in multi-head attention.

**Mamba architecture** Mamba 1 and 2 are prominent sequence modeling architectures that combine selective state space layers (as the sequence mixer) with more standard layers. We describe below the Mamba 1 block, and refer the reader to (Dao & Gu, 2024) for details on Mamba 2, which are not essential to our work. Omitting the final LayerNorm, the Mamba block is a composition of two sequence mixer layers (1D convolution and a selective SSM layer) a gated linear block:

$$W_3\{\mathsf{SSM}\left[\sigma(\mathsf{DepthwiseConv1d}(W_1 X))\right] \odot \sigma(W_2 X)\} + X, \tag{9}$$

where $\sigma$ is SiLU (Hendrycks & Gimpel, 2016). Mamba 2 simplifies this block, merging all projections into $W_1$. For both, the convolution layer before the SSM will be considered in our initialization.

**Mamba attention maps** Throughout this work, we visually inspect $M_d$ to better understand the operation implemented by Mamba layers. However, it is infeasible to look at all D maps, and we instead visualize and report the average over channels $\frac{1}{\text{D}}\sum_{d=1}^{\text{D}} M_d$, which we hereafter refer to as the *attention map* of a Mamba layer. In practice, the inter-channel variation in maps is relatively small, as the behavior of $M_d$ is dominated by $\bar{B}_d$ and $C$. We also sometimes find it useful to inspect the *average attention mask* $\frac{1}{\text{DN}}\sum_{d=1}^{\text{D}}\sum_{n=1}^{\text{N}}(\Pi_{k=j+1}^{i}\text{diag}(\bar{A}_{d,:,k}))_{n,n}$ to approximately determine the effective receptive field of the Mamba layer (i.e., how far into the past it can look).

**Copying task** Most of our experiments focus on copying, a simple task where SSMs are known to fall far behind Transformers. We train the model to predict the paste string given the copy string, emitting a stop token at completion.

$$\underbrace{\texttt{abcdefghijk}}_{\text{copy string}} \quad | \quad \underbrace{\texttt{abcde}\,\underset{\cdot}{?}}_{\text{paste string}} \cdots \square \tag{10}$$

Since Transformers cache the whole sequence, it is easy for them to learn the task and to generalize far beyond the training length. However, since SSMs compress tokens into a fixed-size state, it is hard for them to store and decode back long sequences. We consider copying sequences of varying length and of different vocabulary size, drawing tokens uniformly at random. We also investigate *stack-order* copying, where the paste string needs to be generated in the reverse order.

**Multi-query associative recall** Another synthetic task that has been shown to be an important discriminator between Transformer and SSM abilities is multi-query associative recall, which tests models' ability to store and recall many key-value pairs. Transformers are well-suited for this task, as they can implement induction heads easily (Olsson et al., 2022).

$$\underbrace{\texttt{a1 b2 c3 d4}}_{\text{key}-\text{value pairs}} \quad | \quad \underbrace{\texttt{c3 b}\,\underset{\cdot}{?}}_{\text{queries}} \cdots \square \tag{11}$$

Similarly to copying, we investigate length generalization on multi-query associative recall. In our implementation, each key may occur only once, i.e., it cannot be overwritten by later key/value pairs.

## 3 INITIALIZING STATE SPACE LAYERS TO BE MORE LIKE ATTENTION

To better understand why Mamba often fails to learn to copy, we start by examining a small model trained to copy 50-character strings. In Figure 1a, we can see that Mamba plateaus. Visual inspection of its attention maps reveals that it has probably failed to learn an interpretable copying operation.

**Attention enables copying** To explore what Mamba might be missing to allow it to copy, we trained a hybrid eight-layer Mamba whose fourth layer is single-head self-attention. As shown in Fig. 2, this one layer enables perfect copying performance, both on in-distribution length-50 strings (dotted lines) and generalizing to length-100 strings (solid lines). The softmax attention head learns a sharp "look-behind" operation, constructing the paste string by directly attending to the copy string, likely exploiting an implicit position embedding learned by the preceding Mamba layers. We propose two initialization changes that allow state space layers to better use their state capacity.

**1. State space layers can be linear attention** While there is likely more than one way to learn to copy, we suspected that Mamba's copying ability is tied to its ability to represent a similar operation to the one in this self-attention layer. Notably, in Figure 1a, the Mamba layers tend to look only into the recent past, while the self-attention layer in Figure 2 can attend all the way to the beginning of the string. While SSMs cannot look arbitrarily far into the past because of their fixed state size, even in the simplest time-invariant SSMs, the amount of history stored in the state is controlled by the parameter $A$, whose initialization was crucial to the initial success of these models (Gu et al., 2021).

Consequently, we focus on the state matrix $A$, which controls the "receptive field" of the state space layer. Note in Eq. 8 that if $\bar{A}_d \approx 1$, then $M_{d,i,j} \approx C_{:,i}^T \bar{B}_{d,:,j}$. That is, the state space layer's attention map resembles a product of *queries* and *keys*. The only inter-channel variation in this equation is from $\Delta_d$ in Eq. 5, so that if $\Delta_d \approx 1$ then $\bar{B}_d \approx W_B X$, which results in $M = X^T W_C^T W_B X$, which is simple linear attention before applying the causal mask. Thus, if we set parameters so

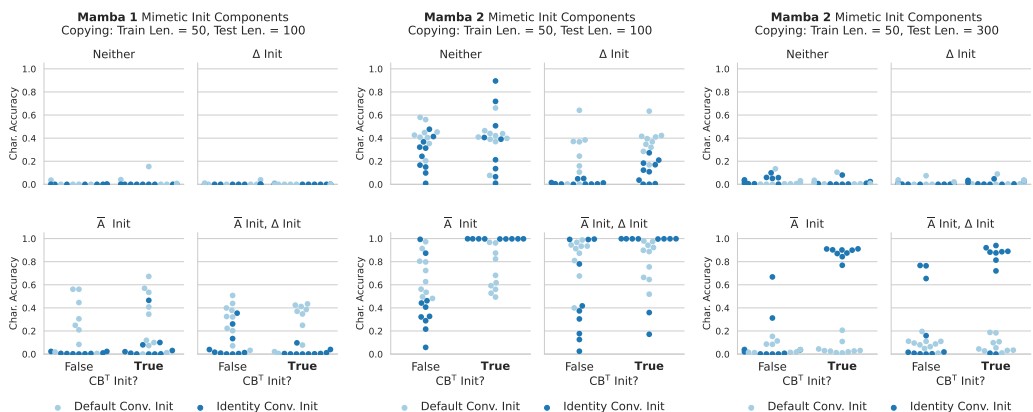

Figure 3: Testing the four components of our initialization on Mamba 1 & 2 for 10 seeds.

that $\Delta_d, \bar{A}_d = 1$, the state space transformation is the same for every channel, and it is simple (non-kernelized) linear attention with head dimension $\texttt{N}$ and no value/projection matrices:

$$\Delta_d, \bar{A}_d \approx 1 \quad \Longrightarrow \quad Y \approx X \cdot \texttt{tril}\left(X^T W_C^T W_B X\right) \in \mathbb{R}^{\texttt{D} \times \texttt{T}}. \tag{12}$$

However, both $\bar{A}_d$ and $\Delta_d$ are parameterized and input-dependent, so we cannot directly set them to one. We use details of the Mamba implementation: To make $\bar{A}_d = \exp(A_d^T \Delta_d) \approx 1$, we parameterize $A = -\exp(-cA_{\log})$, which is nearly 0 for large $c$, making $A_d^T \Delta_d \approx 0$ in Eq. 7. We choose $c$ from $\{2, 4, 8\}$. We then set $W_\Delta \approx 0$ and $b_\Delta = \texttt{softplus}^{-1}(1) \approx 0.54$ in Eq. 4 so $\Delta_d \approx 1$. **This makes the state space layer close to its linear attention counterpart at initialization.**

**2. Correlated tokens should attend to each other** Having shown that state space layers can mimic linear attention, we now try to make them mimic attention layers that can copy, such as the one in Fig. 2, which implements a look-behind operation. We focus on a single linear attention/state space layer, *assuming* the layers before it learned a representation amenable to copying. Consider a copying example of length $n$, where we have already copied $k < n$ of the D-dim. tokens past the delimiter $x_\|$ and want to copy the $(k+1)^{\texttt{st}}$ one: $X = (x_1, \cdots, x_n, x_\|, x_1, \cdots, x_k) \in \mathbb{R}^{(n+k+1) \times \texttt{D}}$. We assume that preceding layers $f$ have learned to superimpose a position embedding as follows:

$$f(X) = (x_1 + p_1, \cdots, x_n + p_n, x_\| + p_1, x_1 + p_2, \cdots, x_k + p_{k+1}) = X + P \in \mathbb{R}^{(n+k+1) \times \texttt{D}},$$

so that token with index $k$ in the paste string will attend to token $k+1$ in the copy string because $(x_{i+1} + p_{i+1})^T(x_i + p_{i+1}) > 0$, assuming $x_{i+1}^T x_i, x_j^T p_j \approx 0$ (uncorrelated) and $p_j^T p_j = 1$ (correlated). That is, $f(X)^T f(X) \approx P^T P$ will have similar structure to that in Fig. 2. In this case, copying behavior will arise in our state space/linear attention layer if $P^T W_C^T W_B P \approx P^T P$, i.e., when $W_C^T W_B \approx I$. Since $W_C, W_B$ are low rank ($\texttt{N} < \texttt{D}$), their product cannot be exactly the identity; using the fact that random Gaussian matrices are semi-orthogonal, we could set $W_C := W_B$ to get $W_C^T W_B \approx I$. Initializing the queries and keys to be correlated was also noted by Trockman & Kolter (2023), who suggest these weights should not be strictly equal, so we instead set $W_C := \frac{1}{2}(W_C' + W_B)$. In summary, assuming the model has learned a useful correlation structure between tokens, setting $W_C^T W_B \approx I$ ensures this structure can be leveraged by attention. For similar reasons, we experiment with initializing the convolution in Mamba layers to the identity.

**Which of these components matter?** In Fig. 3, we investigate the interaction of these four possible mimetic initialization components, displaying all sixteen possible off/on combinations. We investigate copying on 50-long strings and generalizing to 100- and 300-long strings for a 24-layer Mamba with hidden size 1024 as in Jelassi et al. (2024). For the $A$ and $\Delta$ initializations, we fix $c = 8$ and $b_\Delta = 0.54$. For Mamba 1, we

| Initialization | Purpose |
|---|---|
| $A \approx 1$ $\Delta \approx 1$ | Approximate linear attn |
| $W_C^T W_B \approx I$ $\texttt{Conv1d} \approx I$ | Encourage recall |

see that there is only a significant effect when setting $A \approx 1$, with no apparent benefit to setting $\Delta \approx 1$; while setting $W_C^T W_B \approx 1$ has only a tiny effect, using identity convolution initialization seems somewhat harmful.

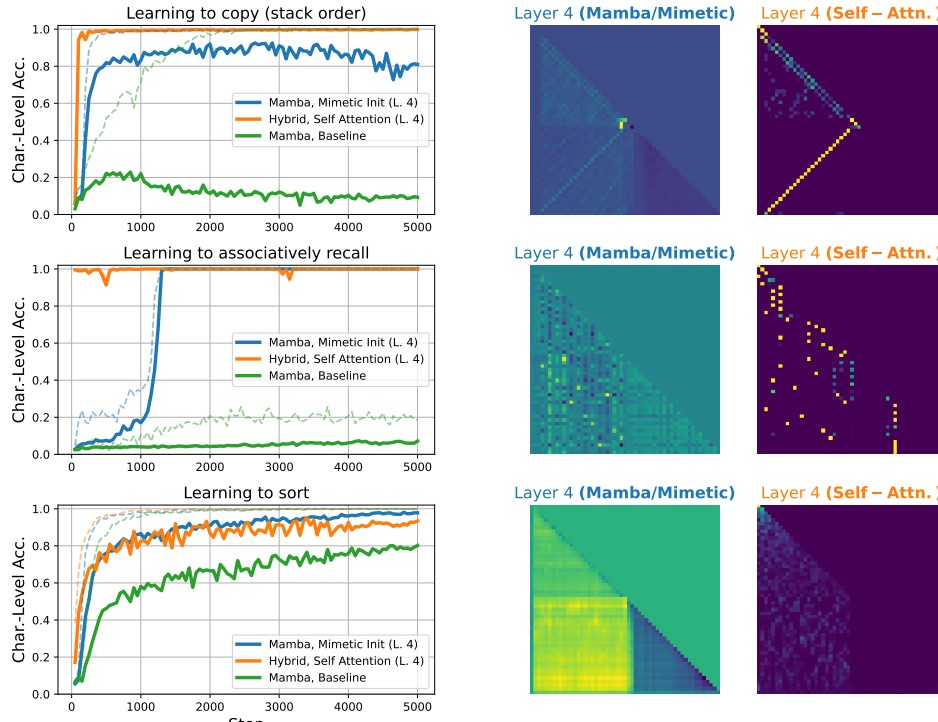

Figure 4: Mimetically initialized Mamba layers learn similar operations to Self-Attention layers in the same location *naturally* with no additional supervision on several tasks. Dotted lines: accuracy at training length (50), solid lines: generalizing to length 100.

For Mamba 2, we see a similar advantage to using $A \approx 1$ initialization, and a advantage to $W_C^T W_B \approx 1$ even without $A \approx 1$, and the two interact to create even better models. Adding identity convolution initialization leads to much better performance still, reaching 100% accuracy in many cases. The positive interaction between $A \approx 1$ and $W_C^T W_B \approx 1$ and identity convolution is especially apparent for 300-long strings.

The difference in the best initialization strategy for the two architectures is likely explained by the removal of linear blocks after the convolutional layer in Mamba 2, as well as the addition of multiple state space heads. Unless otherwise noted, we use the observations above to determine our initialization strategy depending on the Mamba version: For Mamba 1, we use $A, \Delta \approx 1, W_C^T W_B \approx I$, and for Mamba 2 we add identity convolution initialization.

## 4 STATE SPACE MODELS WANT TO BE TRANSFORMERS: MIMETIC INITIALIZATION LETS THEM GET CLOSER

Mimetic initialization leads to immediate and significant improvements in copying ability. In Fig. 1b, we can see that mimetic initialization allows a small 4-layer Mamba to learn to copy strings with twice the training length with reasonable accuracy in just a few hundred steps, which is far better than the tens of thousands of steps reported in previous work (Jelassi et al., 2024). Note that mimetic initialization leads to Mamba learning a state space layer whose attention map replicates the structure of that of self-attention in Fig 2; i.e., this layer has learned to (continue to) implement linear attention. Mimetic initialization allows Mamba to quickly learn to copy from scratch.

**One mimetic init is all you need?** We continue our investigation of using mimetic initialization to help Mamba learn recall tasks: Given our observations that a single self-attention layer is sufficient to learn these tasks to high fidelity, and that a single Mamba layer can roughly approximate this attention, we use mimetic init for *just one layer* in the same position (Layer 4) of an 8-layer Mamba.

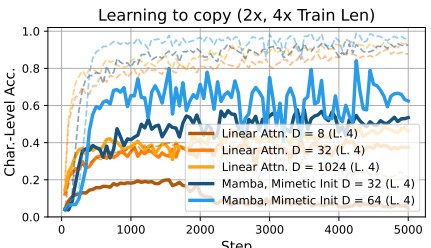 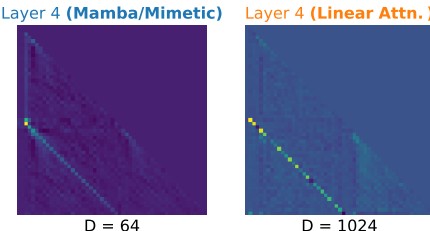

Figure 5: Simple linear attention underperforms Mamba even for very high head dimension, especially at generalization. Dotted lines: accuracy at length 100, solid: at length 200; train length: 50.

In addition to copying (Fig. 2), we present results for additional three synthetic tasks in Fig. 4. First, we investigate copying in stack order, as unpacking the compressed string in most-recently-added order is potentially easier for SSMs. Unlike normal copying, baseline Mamba is able to fit to the training length, but it fails to generalize. Mamba with mimetic init fits the training length much faster and generalizes better, while the self-attention hybrid generalizes nearly immediately. The story is similar for multi-query associative recall – mimetic initialization leads to rapid learning and generalization to twice the length. We also consider the sorting task, where tokens are sampled without replacement from a vocab of size 512. Surprisingly, Mamba with mimetic init does even better than self-attention. Mimetic initialization results in large improvements for all synthetic tasks considered.

**Is Mamba with mimetic init *just* linear attention?**   In Figures 2 & 4, notice that the mimetic initialized Mamba layer tends to mimic the corresponding self-attention layer in the hybrid model; the resemblance is clear for copying in normal and stack order. For associative recall, it is less clear, but the Mamba layer looks significantly more like it could implement a induction-head-like function than typical Mamba layers. Similarly, the interpretation is unclear for sorting, but the overall structure matches. At a high level, it seems like Mamba attempts to learn an approximation to self-attention, but has much less capacity and sharpness. Consequently, we ask if our initialization merely turns state space layers into single-head linear attention layers.

In Figure 5, we present an ablation study where we replace the target Mamba layer in our copying experiment with simple causal linear attention with various head dimensions. According to Eq. 12, we may expect mimetic init to make Mamba layers equivalent to unkernelized linear attention layers with head dimension equal to the state dimension. Consequently, we compare Mamba with state size 32 to linear attention with head dimension 32, which comes relatively close. We plot generalization to $2\times$ and $4\times$-length in Fig. 5, as the difference for fitting to the training length is small. Nonetheless, Mamba still performs somewhat better than linear attention. Linear attention performance depends on the head dimension, with dimension 8 severely underperforming Mamba and dimension 1024 barely exceeding the performance of 32. In contrast, doubling the state dimension of Mamba to 64 substantially improves generalization performance. We visualize the difference in attention maps for the two operations; we can see that Mamba's is perhaps sharper/more consistent like that of self-attention. Combined with better performance on copying, we conclude that mimetic init Mamba layers are not *just* linear attention, but rather a related and superior (for this task) non-linear operation. The correlation between this "sharpness" and linear attention performance has been exploited by recent work (Zhang et al.).

## 5   FURTHER EXPERIMENTS ON MIMETIC INITIALIZATION

Mimetic initialization improves the recall abilities of Mamba 1 and 2 over a variety of architecture settings and sequence lengths. For all Mamba 1 experiments, we use state size 32, though we explore different state sizes for Mamba 2, which has state size 128 unless otherwise noted. For Mamba 2, we use head dimension 64 for all experiments. All trials are for 5000 steps unless otherwise noted, and we swept over a small set of learning rates; our training pipeline is taken from Jelassi et al. (2024). *Note:* While mimetic initialization has a strong effect size for Mamba 1, the architecture generally struggles to copy for larger vocab sizes in the training lengths studied, so we present Mamba 2 results for most larger-scale experiments in the paper. Error bars are computed over five seeds.

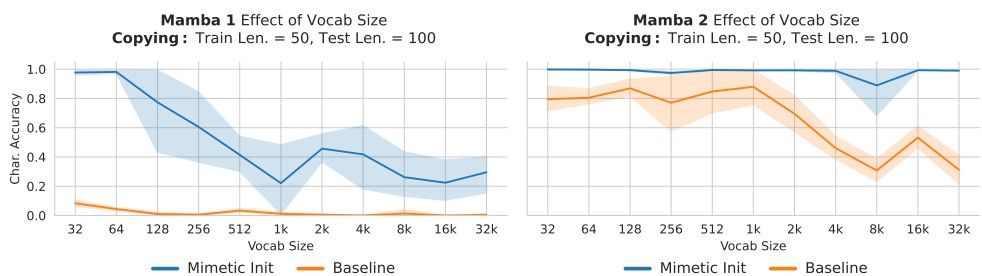

Figure 6: Mamba 2 with mimetic init can learn to copy even for large vocabulary sizes.

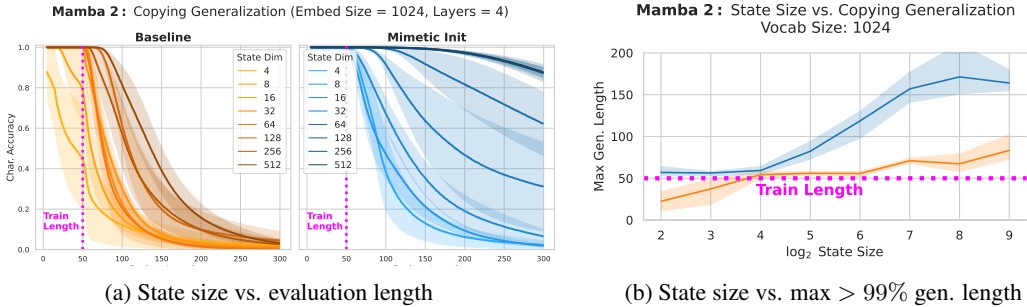

(a) State size vs. evaluation length

(b) State size vs. max $> 99\%$ gen. length

Figure 7: Mimetic initialization allows for better use of the state size for copying; capacity grows roughly linearly with state size, compared to almost not at all with default init.

**Vocabulary sizes**   The larger the vocabulary, the more bits it should take to encode content of a token to enable copying, and the harder it may be to memorize and copy the sequence. While the previous work on copying focused on small vocabularies, we showcase the ability of mimetic init to improve copying even for large vocabularies in Fig. 6. For Mamba 1, mimetic init allows decent copying performance up until a point, and then degrades. In contrast, baseline never learns to generalize. For Mamba 2, mimetic init enables consistent $2\times$ length generalization across sequence lengths, preventing the degradation with vocab size demonstrated by the baseline.

**State dimension**   The copying ability of Mamba should be directly related to its state size, according to Jelassi et al. (2024). This allows Mamba to more easily approximate self-attention-like maps, as we saw earlier. We show this is indeed the case in Fig. 7a. Indeed, for baseline Mamba 2, perfect copying at training length 50 is only possible for sufficiently large state size. However, if we use mimetic initialization, the additional capacity from the state size is much more efficiently used, and generalization (measured with the area under the curve) is far stronger – $N = 32$ with mimetic init achieves performance comparable to $N = 512$ with baseline init, a $16\times$ improvement in the use of capacity. We show another view on this data in Fig. 7b; generalization length hardly grows with the log of the state size using baseline initialization, while it grows linearly only after using mimetic initialization. Mimetic init allows Mamba 2 to get closer to its true compression/copying capacity.

**Architecture size**   In Figure 8, we investigate mimetic init over different Mamba sizes (dimension, layers). Surprisingly, a mere two layers seems to be sufficient, with deeper networks improving generalization beyond $2\times$ length. With embedding size 1024, Mamba 2 can copy very well for a variety of depths; for multi-query associative recall, slightly deeper networks seem preferable. In almost all cases, mimetic initialization leads to superior generalization performance.

**Sequence length**   Mimetic initialization lets us nearly perfectly fit to the training length even for longer strings for both copying and multiquery associative recall (Fig. 9). While baseline tends to struggle to learn to copy even 1000-long strings, mimetic initialization allows fitting to around 4000-long strings. For MQAR, baseline breaks down around 900-long strings, while mimetic initialization allows fitting to 1800-long or more. The benefits apply for better generalization as well, though Mamba still cannot strongly generalize to much longer strings than trained on.

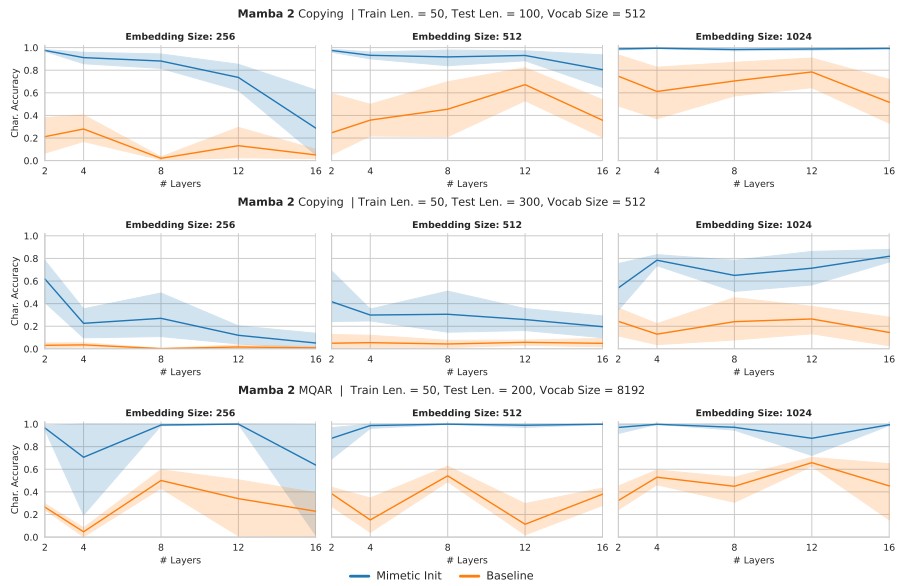

Figure 8: Mimetic initialization vs. Mamba 1/2 architecture sizes.

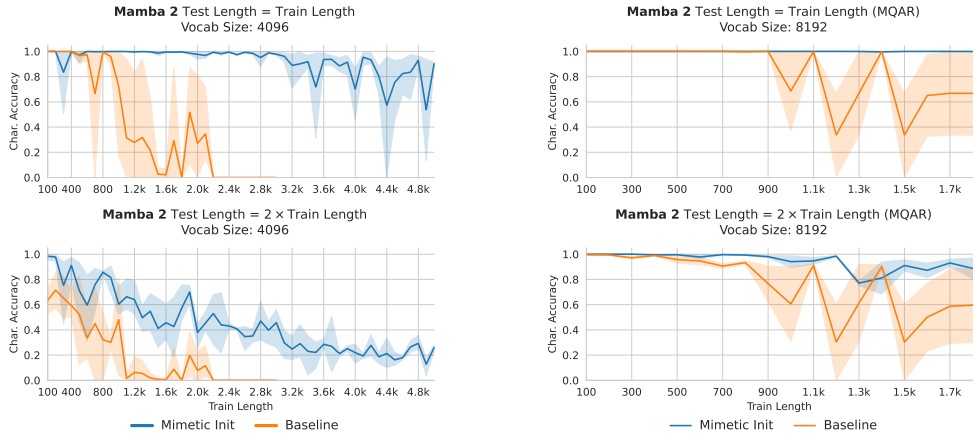

Figure 9: Mimetic init lets us nearly perfectly fit in-distribution even for long sequences on copying (left) and MQAR (right), and also boosts generalization performance (1024-dim 2-layer Mamba 2).

## 6 COMPARING MIMETIC INITIALIZATION TO PRETRAINING

**Mimetic init mimics benefits of pretraining**   We hypothesized that Mamba's difficulty in copying may be an optimization issue rather than fundamental capacity limitations. That is, a Mamba that was first pretrained on a general text corpus may be a better representation of true copying abilities; i.e., one should never train from scratch (Amos et al., 2023). In Fig. 10, we see that finetuning a pretrained 130M Mamba to copy or do associative recall on 50-character strings results in good generalization, but training from scratch with mimetic init achieves similar results. Note that the pretrained Mamba had a much longer ($> 1$k) training length than our from-scratch trials. Considering this, our mimetic init results get impressively close (esp. for shorter strings; dotted lines).

**Localizing the benefit of pretrained weights**   Based on our linear attention observations, the copying abilities of a pretrained Mamba may be localized to a few layers, so we explore the capabilities of individual layers: We use a pretrained teacher Mamba with layers $T_i : i \in [L]$, and then train $L$ student Mambas where each of the $S_j : j \in [M]$ layers is initialized with $S_j := T_i$ for $i \in [L]$. In

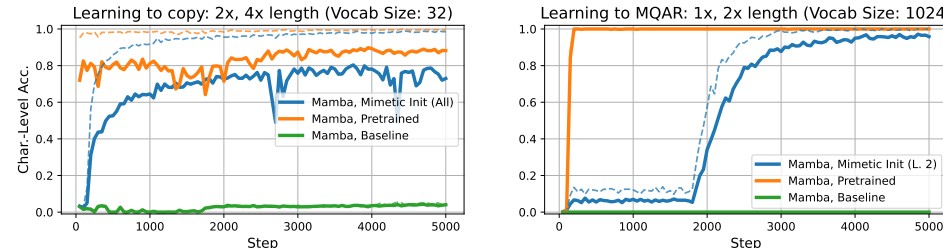

Figure 10: Pretrained 768-dim. 24-layer Mamba 1 vs. from-scratch training (w/ mimetic init).

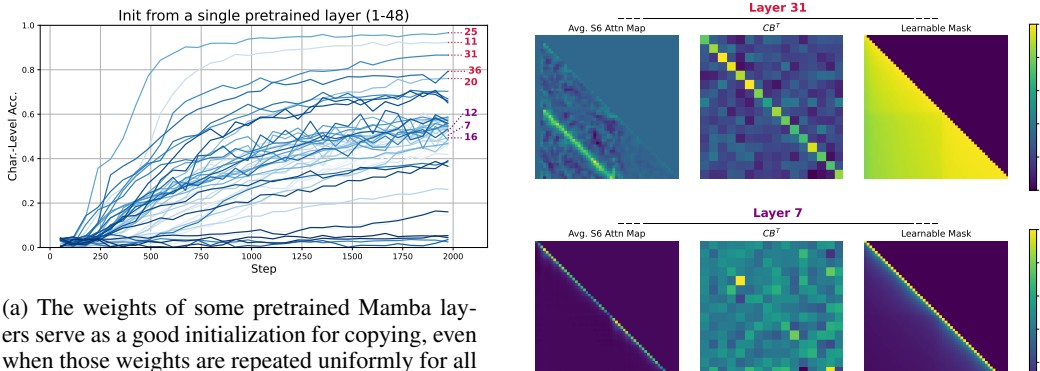

(a) The weights of some pretrained Mamba layers serve as a good initialization for copying, even when those weights are repeated uniformly for all layers in the "student" model. The layers that work well as an initialization for copying tend to have correlated $C$, $B$ weights and nearly-all-ones masks, such as Layer 31.

(b) Some pretrained Mamba layers have structure conducive to copying (Layer 31); others merely mix tokens with their nearby neighbors; from a 26-char test string.

Figure 11: The copying ability of a pretrained Mamba may be attributable to a fraction of its layers.

this case, $L = 48$ and $M = 12$. Using these pretrained weights can make it much easier to learn to copy (Fig. 11a), but the effect size stands out for some particular layers, such as $T_{31}$.

We inspected the weights and attention maps of these layers to see what might be behind the improved performance; see some examples in Fig. 11b. Some layers such as $T_{31}$ look like our mimetic initialized layers, with nearly all-ones average attention masks, correlated $W_C$, $W_B$ weights, and lower diagonal structure, similarly to self-attention layers in hybrid Mambas earlier. That is, the structure our initialization provides seems to arise *naturally* in Mambas trained on sufficiently large and varied corpora, and may be fundamental to Mamba's copying and recall abilities.

## 7 CONCLUSION

We presented mimetic initialization for state space layers, a simple and closed-form technique to greatly improve the copying and recall abilities of state space models. Mimetic initialization makes state space layers mimic linear attention at initialization time, and also mimics the structure of state space layers that contribute to copying and recall abilities in pretrained models. Our technique allows to estimate capabilities of SSMs more accurately, which have been alternatively over- and underestimated in the literature (Jelassi et al., 2024; Waleffe et al., 2024). Using a better initialization such as ours may assist in developing new architectures starting from a smaller scale, allowing for better predictions of their full-scale performance, as is often done in practice in testbeds (Poli et al., 2024). From a theoretical perspective, our particular construction may provide insights into the tradeoffs between state space layers and attention, and may help to study the recall vs. non-recall capabilities of state space layers. Improving the ability of state space layers to approximate attention has already been noted in followup work to the original Mamba architecture (Dao & Gu, 2024), and our initialization supports this concept. More broadly and together with previous work on mimetic initialization, our work helps to better understand pretraining, to some extent disentangling its dual purposes of storing knowledge and serving as a good initialization.

## 8 REPRODUCIBILITY STATEMENT

We have provided all the necessary details to reproduce our findings in the main text. All experiments were done with multiple random seeds, reporting the average and error bars. We swept learning rates over $\{0.001, 0.0005, 0.0001\}$. We used the code from Jelassi et al. (2024), found at `https://github.com/sjelassi/transformers_ssm_copy`, and used pretrained weights from `https://huggingface.co/state-spaces/mamba-130m` and `https://huggingface.co/state-spaces/mamba-370m` in some experiments. For multiquery associative recall, we used code from `https://github.com/HazyResearch/zoology`. On two A100 GPUs, most training runs take around 30-60m to complete, with longer training times for very deep models or those trained on very long sequences. Source code will be released after publication.

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

# A    APPENDIX

