# OpenReview forum: "Mimetic Initialization Helps State Space Models Learn to Recall"
_ICLR.cc/2025/Conference — Submitted to ICLR 2025_

### Official Review · Reviewer_syxM · 2024-10-29

**Soundness:** 3
**Presentation:** 2
**Contribution:** 2
**Rating:** 5
**Confidence:** 4

**Summary:**

This paper addresses problems with copying (and more generally recall) in state-space models by proposing a novel "mimetic initialization" scheme based on the following two observations:

1. **State-space models as a generalization of linear attention**: The trainability of the $A_i$ matrices in state-space models allows them to act as a generalization of linear attention. However, this also can give rise to time decay that is too aggressive to learn copying tasks.
2. **Preserving correlation structure between input tokens**: By initializing the product of the $B$ and $C$ projection matrices to be the identity, the authors aim to maintain the correlation structure between input tokens.

By leveraging these two observations to construct an initialization scheme, the authors demonstrate that the resulting mimetic Mamba 1 and Mamba 2 architectures outperform the default Mamba 1 and Mamba 2 architectures on both copying and MQAR tasks. The authors extend the results across multiple task and model hyperparameters and conclude by demonstrating how mimetic initialization can be interpreted as an efficient alternative to model pretraining.

**Strengths:**

The main strengths of this paper lie in the simplicity of the proposed initializer and the robustness of the empirical results. In particular, I note the following:

1. The empirical findings are strong. Specifically, the mimetic Mamba models achieve significantly higher performance on both copying and recall tasks compared to the default Mamba models. They also reach this performance at a much faster rate.

2. The empirical evaluation is thorough. The authors conduct extensive sweeps across various vocabulary sizes, state sizes, and sequence lengths, demonstrating that mimetic Mamba consistently outperforms default Mamba across these settings.

3. The authors offer insight into how mimetic Mamba functions as a pretraining prior, providing intuition for why the initializer is effective.

4. The work appears reproducible based on the statement provided by the authors, although code has not yet been released.

**Weaknesses:**

The weaknesses in this paper primarily lie in lack of presentation, novelty, breadth, and clarity. In particular, I note the following:

1. The paper is not particularly well-written. It seems rushed, and while the overall story makes sense, it lacks the logical flow both within and between sections that would make it easier for readers to follow. Some examples of this include the related works which would benefit from bolded paragraph headers like the authors use throughout the paper, rewriting the "Correlated tokens should attend to each other" section which is currently quite confusing to parse, moving the figures so they are actually near the text the author is reading, and more generally adding transitions between the sections that motivate why we need to do the next analysis so the story flows better. Also, I would consider moving the pretraining section after the first set of results since it provides nice intuition as to why the initializer works.

2. The idea is not particularly novel compared to "Mimetic Initialization of Self-Attention Layers" (Trockman 2023), which is cited throughout the paper. Specifically, the authors adopt the concept of correlating the query and key projection matrices to maintain correlation structure amongst tokens from this prior work and simply apply it to the analogous components of Mamba.

3. The authors’ use of an identity prior on $A$ aligns with the intuition of orthogonal/unitary non-linear RNNs, a related line of work that is entirely ignored in the paper. Although the focus here is on linear systems, it seems incomplete not to mention this prior work. From my understanding, the approach presented in this work effectively reduces the rate at which the gradient vanishes over time, which is a longstanding idea in RNN research.

4. The authors restrict their results to synthetic tasks and do not demonstrate the applicability of this initializer (or any intuition derived from it) on language tasks, limiting the scope of the findings. Maybe something like the Pile is appropriate here?

5. The authors do not analyze other SSMs outside of Mamba (e.g., GLA, RWKV, Hyena, etc.), despite claiming that this initializer improves performance in SSMs more broadly. Intuitively, mimetic initialization should extend to something like GLA which has a distinct functional form from Mamba in the state-to-state dynamics. This would be nice to show.

6. The authors benchmark this initializer only against the default Mamba initializer, even though the original Mamba paper presents other initializers that could have been included for comparison (these could be included in supplementary materials).

**Questions:**

Given the weaknesses listed above, I will recommend that this paper be rejected. While the proposed initializer is simple, intuitive, and seems effective, it lacks novelty, and the downstream analyses lack breadth. Furthermore, there is little theoretical justification in this paper (most of which appears to be drawn from prior work). This is acceptable only if a varied empirical analysis compensates for it, which is currently lacking. I think this paper would benefit significantly from another iteration. In particular, I would like to see the following addressed:

1. Explain why the line of work on orthogonal RNNs is not relevant here. If it is, please incorporate it into the paper. In particular, a more extensive related work section in the appendix would add substantial value.

2. Figure 2 is confusing. If the dots correspond to 10 random seeds, why not present the plot as an average with error bars? A more thorough multidimensional display could better reveal the meaningful variations across the ablated dimensions discussed later in the text.

3. The legend in Figure 5 is obscuring half of the plot.

4. Producing results that demonstrate utility beyond synthetic tasks would significantly improve the paper.

5. The authors note that Mamba outperforms linear attention on synthetic tasks but do not offer any intuition for this. Additionally, are the authors using the short convolution from the Mamba block in linear attention? If not, how is this a fair comparison? Please clarify

6. Much of the paper builds on linear attention, as this architecture is intended to serve as a prior in the mimetic initializer. If only one other SSM is to be benchmarked against, why choose Mamba? Isn’t GLA (or even S6) a more appropriate architecture for comparison? Mamba’s discretization step and short convolution are acting as unnecessary confounding factors. More generally, this initializer would be stronger if its effectiveness was demonstrated across a broader range of SSMs rather than just Mamba. Even so, could the authors at least please justify their choice of Mamba as the primary comparison model?

7. To what extent do the models deviate from the initialization prior after training?

8. In which regimes is the initializer most effective? Is it with many tokens to copy, long sequence lengths, etc.? Some plots potentially contain these results, but are hard to parse and there is no discussion in the paper.

9. Can the authors clarify the purpose of the MQAR analysis? Is it intended as a supplemental synthetic analysis, or does it show something distinct from copying? The paper’s claims primarily focus on copying, with MQAR briefly mentioned at the end.

---

> ### Author Response · Authors · 2024-11-21
> **Response 1/3**
>
> Thanks for your thoughtful review! We try to address some potential misunderstandings below, and provide a few more experimental results based on your comments.
>
> ### Weaknesses
>
> 1. We'll improve the flow of the paper; the suggestion to move the pretraining section closer to the intro seems reasonable to us, and this makes the connection to previous work on mimetic initialization somewhat more clear.
>
> 2. We disagree. Only one of the four components of our initialization is similar to that from Trockman 2023, and it is quite unprecedented that it works on language. This paper showed strong results on vision but marginal results on language, so our positive result on language using a similar technique is quite novel. Setting the parameter $A$ to make state space layers close to linear attention is novel, to our knowledge.
>
> 3. This is a significant misunderstanding. Setting $A \approx 1$ is not an "identity prior" -- this controls the receptive field of the learnable causal mask. If anything, this makes the layer behave less like the identity. We make no attempt to avoid vanishing/exploding gradients in the sense of previous work on unitary/orthogonal RNNs. Like previous work on mimetic initialization, we do not consider the "signal propagation" perspective at all, but rather the high-level statistical structure of the weights as inspired by pretrained models.
>
> 4. We directly addressed a notable shortcoming of SSMs such as Mamba, which is that they have been shown to be bad at copying/recall [0]. Improving full-scale pretraining results was outside the scope of the paper, which sought to address this one problem with a simple, intuitive solution. The purpose of our paper was primarily scientific rather than application-driven. For example, we contribute to the hypothesis behind mimetic initialization, that the benefit from pretraining comes from storing transferrable knowledge _and_ serving as a good initialization, and that the second component could be isolated in closed-form. We also show that large but constant state sizes endow models with a surprising amount of copy/recall capacity.

---

> > ### Author Response · Authors · 2024-11-21
> > **Response 2/3**
> >
> > 5. Previous work on copying in SSMs has focused on Mamba in particular [0]. The latest RWKV, RWKV-6 as well as GLA fits into the same selective SSM framework studied in our paper, though Hyena does not belong in this class. We get the impression that Mamba's double exponential parameterization gives it an advantage in learning these tasks relative to other SSMs. Extending our technique to other Mamba- or GLA-like works depends only on the parameterization of the gating parameter, and the same lessons from our paper apply straightforwardly. We did some experiments on GLA in particular, which we copy from another review response:
> >
> > We concede that GLA may be a reasonable comparison to put in the paper. GLA is quite similar to the state space layer in Mamba 1 and 2, and the same technique from our paper applies: try to set underlying parameters such that the gating parameter is close to all-1s, making the layer resemble linear attention. The correlated C/B (equivalently, query/key) phenomenon also applies. However, in our experiments, we were not able to train networks using only GLA to copy. More experiment details below:
> >
> > In particular, we replaced the SSD layer inside Mamba 2 with GLA with the same hidden dimension and number of heads. We attempted to set $\alpha_t = \sigma(x_t W_{\alpha 1} W_{\alpha 2} + b_\alpha )$ so that $G_t = \alpha_t 1^T \approx 1$. In the GLA implementation, $\sigma(x) = \exp( \mathsf{LogSigmoid}(x)/16 )$, so we set $W_{\alpha 1} W_{\alpha 2} \approx 0$ and $b_\alpha \approx 10$ to get $\alpha_t \approx 1$. We compare this to the default initialization, training on 50-length strings, using 4-layer models with hidden dim 1024.
> >
> > Model | Acc. (Length 50) | Acc. (Length 200)
> > -------|---------------------|------------------------
> > Baseline Mamba 2		| 100%	 |	39.1%
> > Baseline Mamba 2 (init)	        | 100%	 |	88.2%
> > All GLA               		|  5.0%	 |	4.2%
> > All GLA (init)			| 99.9%	 |	8.10%
> > GLA Layer 2      		| 100%	 |	13.1%
> > GLA Layer 2 (init)		| 100%	 |	86.1%
> >
> > Models using only GLA were not able to learn to generalize on the copying task despite trying several initialization hyperparameters. Our initialization seems to help fit to the training length, at least. We also tried Mamba 2 models where we replaced SSD in only the second layer by GLA (with/without our init), and this performs comparably to Mamba 2 with our init. We note that GLA is actually closer to all-1s using its default init than Mamba 2 due to the particular sigmoid parameterization, so we might expect (at least this component) of our init to have less of an effect.
> >
> > We're unsure why GLA on its own seems to be unable to learn to generalize on copying, but this makes us think that **including it in the paper may be relatively low priority**.
> >
> > Given the similarity of different selective SSM layers and the relative popularity of Mamba (and the fact that it learns to copy more readily than GLA), we think it's reasonable to study Mamba 1 and Mamba 2 exclusively in the paper. We can add some GLA results to the appendix if this is helpful, in addition to referencing this work.
> >
> > 6. We think benchmarking the different inits is not a great idea since our technique doesn't care about the structure of $A$, only about the scale (this is what makes it close to all-1s and thus close to linear attention). And we trust the creators of Mamba chose the best of the inits to use in production, where they evaluated on synthetic tasks like selective copying [4].
> >
> > ### Questions
> >
> > As you said, our initialization is simple and intuitive, even if not completely theoretically rigorous (deep learning is increasingly an empirical science, after all) -- and as mentioned earlier, quite a bit different from previous work on mimetic initialization which largely applied to vision. While there are surely other experiments we could run, we show that our technique is effective for 2 common SSM architectures across a wide range of architecture hyperparameters such as hidden dimension, depth, vocab size, sequence length, state size, and so on. We triangulate the effectiveness of our approach through a first-principles derivation of the relationship to linear attention as well as a study of the structure of weights of pretrained models. We think this cleanly addresses the noted issue that SSMs struggle to learn to copy [0].
> >
> > 1. As explained earlier, work on orthogonal RNNs is almost entirely irrelevant to this line of work, but if this is not obvious we would be glad to add an explanation contrasting our work in related work section.

---

> > > ### Author Response · Authors · 2024-11-21
> > > **Response 3/3**
> > >
> > > 2. It's unclear why we should aggregate the data and thus lose information when the raw data is itself quite illustrative.
> > > This allows us to show how our initialization changes the success rate of learning to copy. For the 24-layer Mambas in the figure, the success rate is not perfect (it's much higher for shallower models). It's unclear how we would display more than four dimensions in one plot (plotting the interaction of all four components is already somewhat cumbersome). If you think it would be helpful, we can plot averages and error bars over lower-opacity data points, but we do find it illustrative to actually present the raw data.
> > >
> > > 3. We'll put the legend below the plot, this is a good point.
> > >
> > > 4. The purpose of our paper was to address the copying problems noted by [0], and we think that an application to general language modeling/pretraining is outside the scope of our study, which was meant to be a scientific study of the capacity of Mamba/SSMs and a further study of the concept of mimetic initialization.
> > >
> > > 5. Yes, we're using the short conv block before the SSM block. We just replaced the state space layer with linear attention. We'll clarify this in the paper. We can also add some more intuition to the paper, but the idea is just that Mamba is more expressive than vanilla linear attention: it has more parameters for gating and discretization. Vanilla linear attention has a fixed causal mask, while Mamba has a learnable one. Just because we initialize to be close to linear attention does not mean that the layer remains linear attention after training.
> > >
> > > 6. We provide some GLA results above, but previous work on studying copying in SSMs focused on Mamba (and we in fact study two different Mamba architectures), and we were unable to attain good copying results using GLA alone. Most likely the particular parameterization of Mamba just happens to make learning this task easier, but there is undeniable similarity with GLA. Note that S6 is actually just the Mamba 1 state space layer, so we did study that. We believe Mamba 1 & 2 to be the most popular SSM architectures and much previous work has focused on them [0,2], most likely because it works particularly well [2] compared to others. Further, if an architecture is in fact the same "class" as Mamba 1/2, then our technique and the accompanying intuition extends very easily. We'll make this clearer in the paper.
> > >
> > >
> > > 7. Quite a lot, see Figure 1: only one layer retains the linear attention prior in this example.
> > >
> > > 8. The init has a large effect size in basically all regimes studied, and the difference is more a matter of which settings Mamba most struggles to copy/recall. For example, it's better for larger vocab sizes because Mamba especially struggles to copy in this setting and our init almost entirely fixes the issue (Fig 6). The most significant effect of our init is to allow Mamba 2's copying capacity to actually grow with the state size, which doesn't happen otherwise (Figure 7b). Without our init, generalization ability is quite limited. And this is for Mamba 2, which is already much better at copying than Mamba 1. For Mamba 1, our init makes all the difference in essentially all settings: without it, learning to copy is just extremely hard/impossible (e.g., see Fig 6, lefthand side).
> > >
> > > 9. Yes, it's simply a different synthetic task than copying which has been studied by the Mamba 2 paper (Fig. 8) [1]. The copying ability of Mamba has been studied in particular by [0], and the MQAR ability by [2], which is largely why we study both. MQAR synthetic performance has been noted to be correlated with larger-scale performance [3]. It's a distinct task, but one of the main synthetic tasks that seems important in the literature.
> > >
> > >
> > > [0] Jelassi et al. "Repeat After Me: Transformers are Better than State Space Models at Copying" (2024).
> > >
> > > [1] Dao & Gu. "Transformers are SSMs: Generalized Models and Efficient Algorithms Through Structured State Space Duality" (2024).
> > >
> > > [2] Arora et al. "Simple linear attention language models balance the recall-throughput tradeoff" (2024).
> > >
> > > [3] Poli et al. "Mechanistic Design and Scaling of Hybrid Architectures" (2024).
> > >
> > > [4] Gu & Dao. "Mamba: Linear-Time Sequence Modeling with Selective State Spaces" (2024).

---

> ### Comment · Reviewer_syxM · 2024-11-22
> **Response to authors**
>
> I thank the authors for their thorough response. However, many of my questions remain unaddressed and others that were failed to resolve my concerns. I will go over some of the major ones below.
>
> > 1. This is a significant misunderstanding. Setting $A \approx 1$ is not an "identity prior" -- this controls the receptive field of the learnable causal mask. If anything, this makes the layer behave less like the identity. We make no attempt to avoid vanishing/exploding gradients in the sense of previous work on unitary/orthogonal RNNs. Like previous work on mimetic initialization, we do not consider the "signal propagation" perspective at all, but rather the high-level statistical structure of the weights as inspired by pretrained models.
>
> To clarify, when I said identity prior I was speaking with respect to the eigenvalues of $A$ all being 1 in which case there is no decay across the time dimension. This is precisely what linear attention does, since it has no decay factor, which is why its receptive field does not decay as a function of backwards time. Analogously, the closer the eigenvalues of $A$ lie to 0, the faster the signal along the time axis vanishes. Can the authors please explain to be why the proposed mimetic initialization scheme cannot being viewed as placing a prior on the network to avoid quickly decaying signals (or gradients) as a function of backwards time? I agree, you are not considering the signal propagation perspective, but I am noting a similarity between the two. Please let me know where my logic fails.
>
> > 2. We concede that GLA may be a reasonable comparison to put in the paper. GLA is quite similar to the state space layer in Mamba 1 and 2, and the same technique from our paper applies: try to set underlying parameters such that the gating parameter is close to all-1s, making the layer resemble linear attention. The correlated C/B (equivalently, query/key) phenomenon also applies. However, in our experiments, we were not able to train networks using only GLA to copy. We're unsure why GLA on its own seems to be unable to learn to generalize on copying, but this makes us think that including it in the paper may be relatively low priority.
>
> This, in my opinion, is not a sufficient rebuttal. The proposed initialization scheme not working on GLA (which the authors have agreed is quite similar to Mamba) raises questions regarding the robustness and generalizability of the approach. Are the authors claiming that mimetic initialization is only effective on Mamba because its initialization scheme is poorly misaligned with linear attention? If so, this is too narrow of a scope to constitute a conference-worthy paper. If the authors were able to show a suite of linear RNN architectures that like Mamba benefit from the linear attention prior, this would be much stronger.
>
> Side note: I believe the reason the GLA architecture aligns with the linear attention prior is due to the normalization factor $\beta = 16$. Reducing the normalization factor would cause it to deviate more from that prior. It is possible that the author's proposed changes to the GLA initializer would affect recall performance in this regime, but this has not been tested
>
> > 3. Yes, we're using the short conv block before the SSM block. We just replaced the state space layer with linear attention. We'll clarify this in the paper.
>
> Firstly, have the authors ran any experiments without the use short conv block? Many SSM-based architectures outside of Mamba do not use this architectural component. But the claim in the paper is that mimetic initialization helps SSMs recall (not that it helps models with SSM + conv blocks learn to recall). Of course, if the authors are restricting the scope of their claims to Mamba (i.e. when they use the term SSM they are really only referring to Mamba which it seems they may be), then this point is moot.
>
> > 4. Note that S6 is actually just the Mamba 1 state space layer, so we did study that.
>
> So the authors did study Mamba without the conv layer (i.e. the SSM layer in isolation)? Or are the authors saying that they studied in indirectly via Mamba?
>
> > 5. Further, if an architecture is in fact the same "class" as Mamba 1/2, then our technique and the accompanying intuition extends very easily. We'll make this clearer in the paper.
>
> I agree that the technique extends readily to most SSMs. My question is not that, but rather if it actually improves performance in other models or if it is specific to Mamba due to Mamba having a "poor" initializer. And if it restricted in scope to Mamba, why does it not improve performance on recall in other models? Is it because initializers in other SSMs already are close to linear attention? Currently, it appears that authors do not have a good understanding of this

---

> > ### Author Response · Authors · 2024-11-24
> > **Response**
> >
> > > Identity prior
> >
> > Setting the structure of the causal mask which is parameterized by $A$ is simply insufficient for controlling the norm/variance of the activations and gradients. Let's think by analogy to attention -- to be clear, our work rests on the fact that there is a parallel between attention and selective SSMs. There is previous work on ensuring attention layers are signal-preserving [1] which is quite involved and controls the structure of the query/key matrices, which would be necessary for a "signal preserving" SSM init (by analogy, the $C$/$B$ weights). As an aside, the authors of [1] propose a baseline of explicitly including a controllable identity term in the attention matrix, value skip-init. This would be unnecessary for SSM layers since there is typically a residual term controlled by the parameter $D$ anyways (not studied in our paper, but it is an architectural component of Mamba).
> >
> > More explicitly, say we wanted the state space layer to have an identity prior (e.g, no mixing across the spatial dimension). Visually speaking, we would want the Mamba attention maps to look like the identity. We can achieve this either with a very small, local receptive field from $A$ (without our init) or with a full receptive field as in our initialization. Please see Layer 3 in Figure 1(a) as a representative example of what Mamba layers "look like" (fairly identity-like in some sense) _without_ our init.
> >
> > We think there might be some confusion between signal propagation, as in the variance/norm of activations or gradients from layer to layer, and the idea of "history" in the state space model. We do in fact want to increase the range of "history", which is controlled by A. But this is separate from the question of signal propagation as typically studied (which we assumed you were referring to due to "identity prior"), in that many settings of the learnable causal mask/history retention could be equally good from the signal propagation perspective. The traditional signal propagation perspective is too reductive to capture this distinction.
> >
> > While we do make the diagonal matrix $A$ near-identity and therefore near-unitary in the init, we don't enforce this constraint throughout training. After training (or just a single step), $A$ is no longer near-identity in most layers. And note that if $A$ is orthogonal/unitary it can be only one matrix: $I$, due to the structure imposed on $A$. Note the weight-tying structure is different in a selective SSM layer than in an RNN.
> >
> > > GLA
> >
> > If you look at the table we posted, the init does help GLA. It's just that GLA on its own doesn't learn to copy (in particular, not to generalize). GLA only learns to copy + generalize when used along with Mamba layers. We can only speculate why GLA doesn't learn to copy in GLA-only architectures here. But Mamba has become so widespread for a reason (it works well), presumably.
> >
> > We agree with your comment about $\beta$.
> >
> > > Short conv block
> >
> > Short conv blocks can't recall beyond a fixed training length. (And it's a _short_ conv, so indeed very limited in terms of recall.)
> >
> > When we say "state space models", we refer to deep architectures that use state space model layers. It's somewhat unfortunate that "state space models" refer both to a precise mathematical object and to a class of deep learning architectures, but we meant the latter. Mamba is not a "state space model" according to the precise mathematical definition, it's a deep neural network that uses SSM-like layers and others, but we believe the field has decided to call such architectures "state space models".
> >
> > There would not be much to study in terms of deep learning if by "state space model" you mean, quite literally, just a single linear recurrence.
> >
> > To be clear, we used short conv with GLA as well.
> >
> > > S6
> >
> > Not sure this makes much sense to us. When we talk about state space models we talk about state space models in deep neural networks, not their standalone capabilities. It was not the purpose of the paper to ablate every other type of layer. Considering the relationship between SSMs and convolutions, it seems reasonable to allow convolutions.
> >
> > > Extension to other models
> >
> > It did help GLA, it's just that GLA is still bad at copying in either case, even though its initialization is admittedly more linear-attention-like than Mamba's. We're not sure why.
> >
> > [1] He et al. "Modifying Self-Attention for Faithful Signal Propagation" (2024).

---

> > > ### Comment · Reviewer_syxM · 2024-11-25
> > > **Response**
> > >
> > > Yes, I'm referring to the idea of "history" in the state space model, which appears to be the source of confusion.
> > >
> > > Overall, while I appreciate the clarifications that the authors have provided, I still think that the paper is not yet ready for publication and would greatly benefit from another iteration (as other reviewers have noted). In particular, I would love to see a more complete understanding of how this initializer extends to other SSMs as opposed to just Mamba. Thus, I maintain my score.

---

### Official Review · Reviewer_ZnsJ · 2024-11-03

**Soundness:** 2
**Presentation:** 3
**Contribution:** 2
**Rating:** 5
**Confidence:** 3

**Summary:**

The paper investigates the challenges faced by State Space Models (SSMs), such as Mamba, on recall-based tasks compared to Transformer architectures. The authors propose an initialization technique that encourages SSM layers to behave more similarly to self-attention. They empirically show that this initialization significantly improves Mamba's ability to learn memory-related tasks.

**Strengths:**

The paper is well-written and addresses a fundamental limitation of SSMs compared to Transformers, which is still not fully understood. The proposed approach of initializing SSM layers to mimic self-attention is well-motivated, and the observed attention patterns suggest notable similarities in the behavior of the two. The authors present extensive experiments across various tasks and architectural configurations, confirming that the initialization generally improves performance. They also provide comparisons with linear attention and pretrained Mamba models.

**Weaknesses:**

While the authors motivate the problem well, I find that they do not spend enough time discussing the broader implications of their work. From a theoretical perspective, the initialization approach is mainly heuristic, and does not provide a fully satisfactory understanding of why SSMs struggle with such tasks. On the empirical side, it is unclear whether the proposed initialization could improve performance on language-based tasks. Additionally, I have some concerns about the robustness and clarity of the presented results (see questions below). The differing behaviors between Mamba and Mamba2 are also somewhat concerning, given that the motivation for the initialization does not distinguish between the two. To sum up, I would like the authors to clarify the main theoretical and empirical takeaways of their work.

**Questions:**

* The plots showing attention patterns should be explained more clearly. What inputs were used to generate these visualizations? How dependent are they on the seed used for training? How do these patterns evolve over time? I think that this last question would be especially interesting given that the authors try to align SSM and self-attention at initialization, and I wonder whether this causes the dynamics to overall behave similarly, or rather that they simply start close and gradually diverge.
* I find the results in Figure 3 strange: first, it seems that the initialization has relatively small effect on Mamba 1(with accuracy going at best up to 50%) while the Figure 2 shows every different behavior (>90%). Am I misunderstanding something? Second, even the results for Mamba 2 seem to vary greatly based on seed, which is quite undesirable and detracts from the conclusions of the paper.
* Why do the authors believe there is such a large difference between Mamba 1 and Mamba 2? They mention this briefly on line 307, but I think this question deserves more attention.
* Similarly, why should Mamba perform better than Linear Attention, given that it was initialized to mimic its behavior?

Minor:
* Line 258: "who suggest these weights should not be strictly equal" — It may be useful to provide some motivation for this.
* Figure 8 mentions Mamba 1/2, but I believe only Mamba 2 is shown.
* l.485: I believe that the authors use a fixed T_i to initialize all S_j (since L and M are different) but this is not clear.

---

> ### Author Response · Authors · 2024-11-20
> **Response 1/2**
>
> Thanks for your review! We try to clarify some things below.
>
> > Why do SSMs struggle with copy/recall, how does the init address this?
>
> We meant for this to be clearer in the paper, but it's largely because the learnable causal mask has very small "effective receptive field" at initialization time (the layers only look back into the relatively recent past), which is addressed by our initialization. As we mention in the paper, initialization has historically been important for the success of SSMs in deep learning.
>
> > Could the initialization improve language-based tasks?
>
> The point of our initialization was to fix a noted shortcoming of SSMs, that they struggle on copying and recall tasks in particular. Improving non-recall/copying performance is outside the scope of the paper. We suspect that our technique could help at larger scale, especially by using the init for just a fraction of the SSM layers or heads in Mamba 2 -- not applying it to all layers would surely prevent any regression on non-recall tasks.
>
> > Broader implications of the work
>
> We tried to highlight the main implications in the introduction: for one, a noted disadvantage of Mamba is that it struggles to learn to copy, and we greatly improve it in this respect through our initialization. It's a vote of confidence that SSMs could be practical for long-range recall tasks, which would greatly speed up inference.
>
> As we mention in the conclusion, our work is related to a hypothesis from previous work on mimetic initialization. Namely, that pretraining can be decomposed into storing transferrable knowledge + serving as a good initialization, and that it may be possible to disentangle the good initialization component. Our paper presents one constructive example of this. Paired with previous work on the topic, we think this contributes to mechanistic understanding of deep learning.
>
> > Why is there such a large difference between Mamba 1 and Mamba 2?
>
> First, note that our init often still results in very large gains for Mamba 2 in addition to Mamba 1. The question of why Mamba 2 generally works better has not been resolved, but was addressed by the authors of Mamba. They speculate that the restricted structure of the SSD layer could be helpful [1]. We think ablating all the architectural differences between the two models is outside the scope of our paper.
>
> > Attention patterns should be explained more clearly
>
> We explain how we visualize the Mamba attention maps in lines 162-168. The technique is the same as that of [0]. The inputs are the incoming activations $X$, as well as the state space parameters $A, B, C, \Delta$. For softmax self attention, we visualize attention maps in the standard way (we just display $\mathsf{softmax}(QK^T)$) -- there is only one head in our hybrids. We do average over "heads" for the Mamba attention maps as it is cumbersome to look at all of them, but the cross-head variance is quite low. We'll try to clarify this in the paper, please let us know if you have particular suggestions.
>
> > How do the maps evolve over time / depend on the seed?
>
> We didn't explicitly study this, though we can say that the results are generally quite robust across trials except for very deep models, where there is more variance (and the initialization greatly increases the _proportion_ of models that successfully learn the task), as in Fig. 3.
>
> We'd assume the Mamba maps align with the self-attention ones fairly rapidly based on the rate of change in accuracy in the plots (the models learn the task rapidly in few steps), which might make this hard to study. We also think it's important to point out that we don't explicitly try to align Mamba layers with self-attention behavior, it's just that those initialized to be closer to linear attention tend to have similar structure, qualitatively, to self-attention maps.

---

> > ### Author Response · Authors · 2024-11-20
> > **Response 2/2**
> >
> > > Strange results in Fig. 3
> >
> > We didn't mean for Fig. 3 to show the _best_ results for the paper, just the relative effects of each of our initialization interventions. It differs from Fig. 2 because in Fig. 3 we present 24-layer (much deeper) models compared to 8- or 4-layer models in many of the other figures. We will make this clearer in the caption. It turns out that shallower models learn the copying task more easily and with less training. We didn't discover this until we had tuned our hyperparameters on deeper models, which we present in Fig. 3. Our hyperparameters may be suboptimal, but nonetheless our results are very significant and robust. Also note that 50% accuracy is still a considerable improvement as the plot shows generalization to 2x longer sequences than the model saw during training. In other words, Fig. 3 column 1 shows that Mamba _cannot_ learn to copy with any generalization ability without our init.
> >
> > The results vary depending on the seed for very deep models, which we found to be harder to train. The spread is smaller for other model sizes. Many figures in the paper report error bars, if you'd like to get a sense of this. It's true that not every trial is great even with the initialization, but on average the initialization has a huge effect size. Check out Fig. 6 and 7 in particular -- our init makes a huge difference in actually leveraging the model's state size.
> >
> > > Why does Mamba perform better than linear attention when it was initialized to mimic its behavior?
> >
> > Mamba is more expressive than naive linear attention due to its learnable causal mask and discretization parameter. Initialization is just that: though the layers may be close to linear attention at initialization, they learn something more powerful.
> >
> > > Minor comments
> >
> > - Though setting $W_Q = W_K$ is an easy way to get the desired correlation structure, in practice it is standard to initialize these to different values for expressivity.
> > - Thanks, we'll fix this.
> > - Ah, we'll specify that $i$ is in fact fixed.
> >
> >
> > [0] Ali et al. "The Hidden Attention of Mamba Models" (2024).
> >
> > [1] https://tridao.me/blog/2024/mamba2-part1-model/

---

> > > ### Comment · Reviewer_ZnsJ · 2024-11-24
> > >
> > > Thank you for your response. I remain perplexed by the results in Figure 3. First, I don't think the authors ever observe that they use a deeper model compared to other figures. The significant difference in performance should be discussed, also because the impact on deeper models is more relevant in practice. Second, it's not just that accuracy is capped at 50% for Mamba 1, but also that the initialization often has no effect at all. The fact that this is not carefully discussed and measured is a concern. For these reasons, I maintain my score.

---

> > > > ### Author Response · Authors · 2024-11-24
> > > > **response**
> > > >
> > > > Note that "not having any effect at all" tends to happen for the trials using the identity conv init (dark blue) but not without (light blue). For that reason we don't use this component for Mamba 1. The light blue trials fail 1/10 or 2/10 times, and note this is failure on *2x length generalization*. This is a clear improvement to the success rate of learning to copy, which is 0 without the init.
> > > >
> > > > Yes, we did not note the deeper model in the figure, but will change that.

---

### Official Review · Reviewer_rVdr · 2024-11-06

**Soundness:** 3
**Presentation:** 3
**Contribution:** 2
**Rating:** 5
**Confidence:** 4

**Summary:**

This paper provides an initialization scheme that allows SSM layers to learn copying tasks faster than the default initialization strategy.  They demonstrate that their mimetic initialization scheme allows SSM architectures to learn recall based tasks quickly.  They test their method on a variety of configurations, changing state size, embedding dimension, sequence length, and vocabulary size.

**Strengths:**

* The paper presents a wide array of experiments that vary state size, vocabulary size, sequence length, and embedding size to validate their results
* The paper is well written and easy to follow
* The paper provides a way to estimate the capabilities of SSM layers on copying tasks without expensive pretraining

**Weaknesses:**

* This paper specifically trains for recall tasks, so it is unclear if this initialization scheme would lead to benefits for pretraining and not lead to performance regressions on non recall tasks
* Since hybrid models with attention layers that have inherent copying abilities combined with SSM layers are starting to become more prominent over pure SSM architectures, it is unclear if the copying abilities of the SSM components are of prime importance
* The paper does not provide any results for pretraining with this initialization scheme

**Questions:**

* Did the authors try pretraining with this initialization scheme?
* Can the authors provide evidence that this initialization scheme would not lead to performance regressions on the basic language modeling tasks or non-recall based benchmarks?

---

> ### Author Response · Authors · 2024-11-20
> **Response**
>
> Thanks for your review, we've answered some questions below:
>
> > Does this lead to regressions on non-recall tasks?
>
> We think this is unlikely, especially because the technique is effective even when we use it for just one layer (see line 321). Further, pretrained models tend to learn a few SSM layers which implement recall abilities (Fig. 11). Taken together, we think this suggests that initializing just a few layers out of, say, 48 or 64 in a large pretrained model would not harm performance on non-recall abilities.
>
>
> > Is this important with the rise of hybrid architectures?
>
> While hybrids are becoming more popular, full-context self-attention layers remain the most expensive component in those architectures. If we could replace them with a recall-specialized SSM, we would reduce inference costs. Consider a hybrid where the SSM layers are merely specialized rather than using full attention.
>
> > No pretraining results
>
> The main purpose of the paper was to understand why SSMs don't learn to copy/recall very well, as in [0]. We effectively diagnose the problem, allowing for better understanding of the capacity of SSMs and their tradeoffs wrt Transformers. The secondary purpose was to investigate the idea of "mimetic initialization", where the effect of pretraining can be viewed as storing transferrable knowledge _and_ serving as a good initialization -- this second effect being something we were able to localize constructively with our technique + analysis of pretrained models. That is to say, the purpose of the paper was largely scientific, rather than attempting to improve production-grade models.
>
> That said, we've tried our technique on a 500M-param Mamba 2 trained on 20B tokens of SlimPajama. We see a small improvement to training and validation perplexity using our technique to initialize a small number of layers. However, we lack enough experimental evidence to endorse this in the paper, and full-scale investigation of our technique for pretraining is outside the intended scope.

---

> ### Comment · Reviewer_rVdr · 2024-11-21
> **Response to authors**
>
> I have read the authors response, as well as the other reviews and their corresponding responses.  Since multiple other reviewers mentioned the lack of evaluations on language tasks, and the lack of analysis on the effect on pre-training, it still remains a strong concern of mine.  While attention layers can be expensive for large context sizes, in most hybrid architectures they are relatively sparse (meaning only a small fraction of the layers are attention layers).  Hybrid architectures have demonstrated to be performant and relevant, so I believe my original point still stands.  Overall, I do not feel comfortable recommending to accept this paper primarily due to the lack of analysis on pre-training and language tasks.  I maintain my score.

---

> > ### Author Response · Authors · 2024-11-22
> > **Response**
> >
> > We disagree that this is necessary for a paper analyzing SSM architectures. We think asking all architecture papers to do pre-training and large-scale language modeling experiments hurts the field. Studying properties of architectures in careful, controlled settings is important to improve our understanding of the models. Copying is an important primitive for all architectures. Currently the field lacks basic understanding of the differences in how different architectures learn, what are their capabilities, what are the optimization challenges, and so on. While achieving SOTA pre-training is an important direction, asking all papers to do that is detrimental to the field in the long run. So we ask you to reconsider your score and judge the paper based on whether you think copying/recall as studied in this paper is an important primitive for SSMs to perform well on. Earlier work shows that the bad performance of SSMs on copying is due to limited state size, while ours shows that much of the bad performance is due to optimization difficulties, and the large state sizes used in practice can actually allow to copy quite long strings. This can potentially lead to an in depth study of optimization dynamics of SSMs which are quite different from those of Transformers. We hope to encourage and enable these different lines of work even though it isn't explicitly about pretraining and LLMs, but rather about more basic scientific understanding of different deep architectures.
> >
> > It's also true that hybrids often use few attention layers, but even then full attention layers will dominate the inference compute requirements for very long-context modeling. Full softmax attention is undeniably powerful, but this doesn't mean that we should not explore alternatives. We think it would hinder long-term research to simply assume that full attention is the only option and that alternatives do not need to be explored, which is one of the reasons why we are interested in understanding the recall abilities of SSM layers.
> >
> > For what it's worth, we'd also like to mention that small-scale performance on synthetic tasks such as copying and associative recall is correlated with performance at larger scale [0]; these metrics have been used to guide architecture search that is applicable to large-scale pretraining, and we show that better initialization is crucial to correctly measuring performance on these tasks.
> >
> > [0] Poli et al. "Mechanistic Design and Scaling of Hybrid Architectures" (2024).

---

### Official Review · Reviewer_ieed · 2024-11-09

**Soundness:** 2
**Presentation:** 2
**Contribution:** 2
**Rating:** 3
**Confidence:** 5

**Summary:**

The paper discusses the problem of improving recall/copying performance of State Space Models (Mamba). In particular, the paper claims that the difficulties in recalling from distant past instants of Mamba-like models are not due to the finiteness of their state and instead are due to poor initialization of their state transition maps. The main contribution of the paper is a novel structured initialization that allows state space layers to improve their recall capabilities.

**Strengths:**

The paper addresses an important and timely question on how to improve recall capabilities of modern State Space Models (SSMs). The proposed initialization is thoroughly ablated with many toy experiments designed to test recall/copy abilities.

**Weaknesses:**

While the proposed initialization scheme is sound, the novelty of the idea and technical contribution is incremental. Furthermore, the scope of the work is only limited to Mamba-like models and it is not generic to State Space Models (as stated in the title). To increase the scope of this work, given the relatively small scale experiments conducted, I’d suggest the authors extend their results to other SSM models like: GLA [5], LRU [4] and RetNet [8]. If only Mamba-like models are studied it would be good to modify the title since at the moment it reads as if it is applicable to “State Space Models” in general.

The current manuscript is sometimes confused, SSM layers cannot mimic full Attention since the state of an SSM layer is fixed at the outset before seeing any sample, while the Attention’s state progressively increases as more and more samples are processed. This is a fundamental difference that cannot be bridged by keeping the same functional form of the SSM and only changing its initialization. Based on the results in the paper the proposed initialization avoids the collapse of the receptive field of the SSM’s state rather than mimicking Attention.

Some important related references are missing, please see the Questions section below, can the authors discuss them? In particular, it is worth adding an explicit comparison with the analysis conducted in [4] which would help the reader contextualize the contribution of this work in the landscape of modern SSM models.

**Questions:**

1. Line 90, Why does valinna linear attention improve recall? This is not what has been observed in previous works and it is one of the main reasons for the relatively low adoption of vanilla linear Attention in modern architectures (linear Attention variants have been proposed since 2020, see [1, 2]). For example, the lack of recall capabilities of vanilla linear attention is one of the main motivations in [3] (this work is already mentioned in the manuscript).
2. The paragraph starting in Line 354 is confusing (especially the sentence starting in Line 365). It is well know that Mamba layers are more expressive than Linear Attention as the latter is a special case of the former obtained when the state transition map does not fade (i.e. A=1), see for example [5, 6].
3. Line 202. What do the authors mean by “likely exploiting an implicit position embedding learned by the preceding Mamba layers”? Can they be more precise and point to an experiment showcasing that the Mamba layer learns positional encodings implicitly?
4. Line 260. In past works [6, 7] it has been shown that such convolution is critical to further expand the state dimension of the SSM layer and that removing it could cause serious recall issues. Setting it to the identity (i.e. removing it) does not seem to be a good initialization in general. As this manuscript also mentions in Line 269. Do the authors have a theoretical motivation for the difference between Mamba 1 and Mamba 2 when it comes to the effects of the conv1d layer?
5. The proposed method does not help for large scale experiments as the current initialization strategy is learned throughout training (line 520). Have the authors tested their initialization on larger scale language pre-training experiments? Do they expect their initialization to help when models are learned on larger scale experiments that are not only focused on copy/recall toy tasks?
6. Can the authors report the size of the state of the SSM models they studied in the paper?


*Minor:*
Line 36 and Line 107, the Mamba layers are input dependent linear transformations, not time-dependent.


[1] K. Choromanski et al., “Rethinking attention with performers”, arXiv preprint arXiv:2009.14794 (2020).

[2] S. Wang et al., “Linformer: Self-attention with linear complexity”, arXiv preprint arXiv:2006.04768 (2020).

[3] S. Arora et al., “Simple linear attention language models balance the recall-throughput tradeoff”, arXiv preprint arXiv:2402.18668 (2024).

[4] A. Orvieto et al., “Resurrecting Recurrent Neural Networks for Long Sequences”, PMLR 202:26670-26698, 2023.

[5] S. Yang et al., “Gated Linear Attention Transformers with Hardware-Efficient Training”, arXiv preprint arXiv:2312.06635 (2023).

[6] L. Zancato et al., “B'MOJO: Hybrid State Space Realizations of Foundation Models with Eidetic and Fading Memory”. arXiv preprint arXiv:2407.06324 (2024).

[7] S. Yang et al., “Parallelizing linear transformers with the delta rule over sequence length.” arXiv preprint arXiv:2406.06484, 2024

[8] Y. Sun et al., “Retentive Network: A Successor to Transformer for Large Language Models”, arXiv preprint arXiv:2307.08621, 2023

---

> ### Author Response · Authors · 2024-11-20
> **Response 1/2**
>
> We're willing to change the title to reflect the fact that our paper focuses on Mamba SSMs in particular if you think this is a significant improvement. But we'd like to clarify some potential misunderstandings first.
>
> The paper is mostly about the copying task, which previous work has noted state space models struggle on in particular [0]. Consequently, we focus on SSMs which can actually perform on the copying task. This _excludes_ LRU and RetNet: they are *linear time invariant* and _cannot_ generalize on the copying task. *Selective* SSMs like Mamba 1, Mamba 2, and GLA implement an _input dependent_ operation that can actually generalize on copying tasks.
>
> Our usage of "State Space Models" is in line with previous work [0,1], assuming we're restricted to SSMs which _can_ copy and recall (beyond one fixed sequence length).
>
> We concede that GLA may be a reasonable comparison to put in the paper. GLA is quite similar to the state space layer in Mamba 1 and 2, and the same technique from our paper applies: try to set underlying parameters such that the gating parameter is close to all-1s, making the layer resemble linear attention. The correlated C/B (equivalently, query/key) phenomenon also applies. However, in our experiments, we were not able to train networks using only GLA to copy. More experiment details below:
>
> In particular, we replaced the SSD layer inside Mamba 2 with GLA with the same hidden dimension and number of heads. We attempted to set $\alpha_t = \sigma(x_t W_{\alpha 1} W_{\alpha 2} + b_\alpha )$ so that $G_t = \alpha_t^T 1 \approx 1$. In the GLA implementation, $\sigma(x) = \exp( \mathsf{LogSigmoid}(x)/16 )$, so we set $W_{\alpha 1} W_{\alpha 2} \approx 0$ and $b_\alpha \approx 10$ to get $\alpha_t \approx 1$. We compare this to the default initialization, training on 50-length strings, using 4-layer models with hidden dim 1024.
>
> |Model | Acc. (Length 50) | Acc. (Length 200)|
> |--------|--------------------|-----------------------|
> |Baseline Mamba 2		| 100%	 |	39.1% |
> |Baseline Mamba 2 (init)	        | 100%	 |	88.2% |
> |All GLA               		|  5.0%	 |	4.2% |
> |All GLA (init)			| 99.9%	 |	8.10%      |
> |GLA Layer 2      		| 100%	 |	13.1% |
> |GLA Layer 2 (init)		| 100%	 |	86.1% |
>
> Models using only GLA were not able to learn to generalize on the copying task despite trying several initialization hyperparameters. Our initialization seems to help fit to the training length, at least. We also tried Mamba 2 models where we replaced SSD in only the second layer by GLA (with/without our init), and this performs comparably to Mamba 2 with our init. We note that GLA is actually closer to all-1s using its default init than Mamba 2 due to the particular sigmoid parameterization, so we might expect (at least this component) of our init to have less of an effect.
>
> We're unsure why GLA on its own seems to be unable to learn to generalize on copying, but this makes us think that including it in the paper may be relatively low priority.
>
> Re: *"SSM layers cannot mimic full Attention"*. There's a misunderstanding here; selective SSMs such as Mamba 2 can in fact mimic (linear) attention, and this is in fact noted in the title of the Mamba 2 paper "Transformers are SSMs" [1]. We show this in our paper mathematically. Of course, due to the limited state size and lack of softmax, we can't mimic full (softmax) attention.
>
> We can add some references, though we already cite [3] and the time invariant SSM and linear attention papers are not too relevant (as we use no specialized linear attention techniques). [5,7,8] are good references to add though. Let us know if you have further thoughts.

---

> > ### Author Response · Authors · 2024-11-20
> > **Response 2/2**
> >
> > ### Questions
> >
> > 1. Vanilla linear attention _does_ improve recall, see Fig. 3 in the Arora et al. paper you cited. This applies to state space/linear attention hybrid models. Performer (Choromanski et al.) is unusually bad at improving recall among the linear attention variants they propose. Also, **importantly** we are merely making the state space layers closer to vanilla linear attention: they **are not** exactly linear attention after training (some layers end up closer than others, but state space layers are fundamentally more expressive than linear attention with the identity kernel).
> >
> > 2. We'll clear up this section. We meant to explain that our init doesn't cause SSM layers to collapse to "pure" linear attention. We’ll state the purpose of the section in the first sentence, and make it clear that we replace the state space sublayer with linear attention and include the short conv before it.
> >
> > 3. See the ansatz in Section 3, paragraph labeled "2. Correlated tokens...". We think that conclusively showing the particular type of position embedding Mamba layers learn could be an entire paper in itself and is outside our scope.
> >
> > 4. Please note that initializing the conv to the identity *doesn't remove it!* Identity initializations have long been known to be effective [2, 3]. The intuition is that we may not want to blend position info together. More heuristically, the amount of spatial mixing in the conv layer may determine if the SSM copies using ngram lookups or position lookups, and Mamba 1 or 2 may be better suited to one or the other. We think a study of the effects of the conv layer in SSMs deserves its own paper.
> >
> > 5. We don't think you can conclude from this experiment that our method won't help large-scale pretraining. However, it's very hard to determine if our technique will benefit this setting with confidence. The purpose of our paper is to address the failure mode of SSMs studied by [0], rather than to improve pretraining. We have seen a small improvement using our technique for a couple layers in a 500M Mamba 2 trained on 20B tokens, but we lack enough experimental data to endorse this method in a publication. In contrast, our effect sizes on synthetic tasks are extremely large and relatively consistent.
> >
> > 6. See lines 372-374: we use state size 32 for Mamba 1 and 128 for Mamba 2 unless otherwise noted (e.g., in state size experiments).
> >
> > [0] Jelassi et al. "Repeat After Me: Transformers are Better than State Space Models at Copying" (2024).
> >
> > [1] Dao & Gu. "Transformers are SSMs: Generalized Models and Efficient Algorithms Through Structured State Space Duality" (2024).
> >
> > [2] Zagoruyko et al. "DiracNets: Training Very Deep Neural Networks Without Skip-Connections" (2018).
> >
> > [3] Martens et al. "Rapid training of deep neural networks without skip connections or normalization layers using Deep Kernel Shaping"

---

> > > ### Comment · Reviewer_ieed · 2024-11-25
> > > **Response to Authors**
> > >
> > > I thank the authors for their answers and for conducting the GLA experiments, which I find very intriguing and deserving of attention. The fact that GLA performs less effectively than the previous Mamba experiments, despite being so similar to Mamba 2, is precisely what makes it so interesting. While the proposed method is designed to be agnostic to the specific SSM used, the limited success with GLA—and the uncertainty around how much improvement to expect—suggest that a deeper understanding of the underlying problem is needed.
> > > For instance, what conditions on the state dimension are necessary to solve a given copying or recall-oriented task? Clearly, a larger hidden state is advantageous, as it allows the SSM to retain more information and reduces the likelihood of forgetting. This ties back to my earlier comments regarding some vague assertions in the paper, such as the claim that 'SSM layers can mimic full Attention.' These must be clearly defined in the problem formulation and addressed with targeted experimental validation.
> > >
> > > Despite the clarifications provided, I still believe the manuscript is not yet ready for publication and would benefit from further revision.

---

### Meta-Review · Area_Chair_j31W · 2024-12-19

**Metareview:**

This paper studies the recall and copying performance of the MAMBA model to understand why it underperforms compared to transformer-based models. In turn, the authors propose a new initialization scheme aimed at improving recall performance.

The reviewers acknowledge that the paper addresses an important and not yet well-understood problem. The proposed initialization scheme is simple to implement and appears robust, and the paper is well organized and clearly written.

However, the experimental evaluation is limited, and one reviewer found some of the results puzzling. There are also concerns about the paper’s relevance given the rise of hybrid models in the field. (These concerns can be addressed, though.)

While the work is interesting, most reviewers feel it is not yet ready for publication. Although the authors provided additional results during the rebuttal phase, only minor revisions were made to the manuscript. In light of this, I agree with the reviewers that the paper is not ready for publication in its current form. I therefore recommend rejecting this submission, and suggest to resubmit a carefully revised manuscript to another venue.

**Additional Comments On Reviewer Discussion:**

The authors and reviewers engaged in an interesting discussion during the rebuttal period. Additional results have been presented, however, not all concerns were addressed. The authors didn't provide a careful revised version of the manuscript. Overall, the reviewers remained to have reservations after the rebuttal period.

---

### Decision · Program_Chairs · 2025-01-22

Reject